# Shape-Memory Composites Based on Ionic Elastomers

**DOI:** 10.3390/polym14061230

**Published:** 2022-03-18

**Authors:** Antonio González-Jiménez, Pilar Bernal-Ortega, Fernando M. Salamanca, Juan L. Valentin

**Affiliations:** 1Materials Science and Engineering Area, Rey Juan Carlos University, C/Tulipán s/n, Móstoles, 28933 Madrid, Spain; 2Department of Elastomer Technology and Engineering, University of Twente, Driener-Iolaan 5, 7522 NB Enschede, The Netherlands; m.d.p.bernalortega@utwente.nl; 3Instituto de Ciencia y Tecnología de Polímeros (CSIC), C/Juan de la Cierva 3, 28006 Madrid, Spain; fms@ictp.cic.es

**Keywords:** shape-memory elastomers, shape-memory effect, smart rubbers, ionic elastomers, XNBR, soft polymers, MWCNT

## Abstract

Shape-memory polymers tend to present rigid behavior at ambient temperature, being unable to deform in this state. To obtain soft shape-memory elastomers, composites based on a commercial rubber crosslinked by both ionic and covalent bonds were developed, as these materials do not lose their elastomeric behavior below their transition (or activation) temperature (using ionic transition for such a purpose). The introduction of fillers, such as carbon black and multiwalled carbon nanotubes (MWCNTs), was studied and compared with the unfilled matrix. By adding contents above 10 phr of MWCNT, shape-memory properties were enhanced by 10%, achieving fixing and recovery ratios above 90% and a faster response. Moreover, by adding these fillers, the conductivity of the materials increased from ~10^−11^ to ~10^−4^ S·cm^−1^, allowing the possibility to activate the shape-memory effect with an electric current, based on the heating of the material by the Joule effect, achieving a fast and clean stimulus requiring only a current source of 50 V.

## 1. Introduction

Smart materials, often known as functional materials, have the capacity to adapt to external environmental conditions and experience physical change. Among them are shape-memory materials, which can recover their original shape from a deformed (temporary) shape after being exposed to various external stimuli [1]. Different families of materials have been used to experiment with this effect. Shape-memory alloys demonstrate great memory properties for some applications but have some limitations regarding their deformation capabilities and density. In this sense, polymeric materials used as shape-memory polymers (SMPs) are interesting due to their low density and great deformation capacity to adapt to different shapes, with elastomers being the most promising material to obtain high deformation capacity [2]. A thermal stimulus is the most common stimulus to activate shape change [3,4], but it causes some application issues, as the heating of materials can be difficult to achieve or slow depending on the final temperature due to the electrical insulation of most polymeric materials. For this reason, research efforts have been made to investigate other activation stimuli, such as light [5], electric [6,7], magnetic fields [8,9], water [10,11], and humidity [12].

There is an increasing demand for soft actuators [13,14], so, in recent years, different approaches have been made to obtain polymers that preserve an elastic nature in their temporary shapes, including an interpenetrating crystallizable thermoplastic network with an elastomeric matrix [15], liquid crystalline elastomers [16], and blends of elastomers and small molecule additives [17]. The most popular approach to create elastic behavior in order to deform material into a different shape is the introduction of a dual network, where at least one of them presents a thermal transition, such as glass transition or melting temperature [18]. However, polymers present rigid behavior at ambient temperature, being unable to deform in this state.

Different routes have been proposed to convert elastomers into these functional materials. Natural rubber has been used by Katzenberg et al. [19] and Heuwers et al. [20] in order to obtain shape-memory properties, where crystallization under deformation was the responsible mechanism for fixing the temporary shape. Pantoja et al. employed another approximation by the introduction of fatty acid salts on an elastomeric natural rubber matrix [21]. In these cases, the final materials exhibited elastomeric behavior at ambient temperature, but they became rigid in their temporary shapes until their original shapes were recovered.

Other approaches include the use of noncovalent supramolecular interactions, such as multiple hydrogen bonding [22], hydrophobic interactions, π–π stacking, metal–ligand coordination [23], and ionic interactions [24]. Among the latter group, different ionomers featuring the shape-memory effect have been reported [25]. Weiss et al. [26,27] designed elastomeric ionomers of sulfonated ethylene propylene diene rubber (SEPDM) with low-molar-mass fatty acids or their salts (e.g., zinc stearate). Xie et al. [28,29] investigated the shape-memory properties of a thermoplastic polymer perfluorosulfonic acid ionomer (PFSA), Nafion™, with ionic clusters in a semicrystalline matrix, and Zheng et al. fabricated rubber composite films, which are fabricated via the film formation of carboxylic styrene butadiene rubber (XSBR) latex and citric acid (CA) solution [30]. Salaeh et al. also introduced the ionic network of a PVDF/XNBR blend by incorporating ZnO [31]. All these materials presented rigid behavior at room temperature, as their shape-memory fixing mechanism was based on crystallization. To solve this, in a previous study conducted by our group, ionic elastomers based on a carboxylated nitrile rubber (XNBR) were obtained, where ionic transition was the switching mechanism of the shape-memory effect and elastomeric behavior was preserved in all ranges of temperature [32]. These materials possess strong, physically crosslinked networks featuring the phase separation of ion-rich domains, where trapped polymer chains around ionic associations act as reinforcing points [33]. Nevertheless, this system presented some limitations. The dynamic nature of ionic crosslinks translates into the loss of mechanical properties at high temperatures. To minimize the loss of temporary shape fixing with time due to stress relaxation mechanisms, a relatively low number of covalent crosslinks (0.5 parts per hundred of rubber (phr) of dicumyl peroxide (DCP) vs. 4 phr of MgO) was added. The presence of the covalent crosslinks enhances shape recovery up to 90%, but it has a negative effect on the recovery ratio of the material (less than 85%), as ionic crosslinks are not completely fixed/frozen in time.

Moreover, although these elastomeric materials have some properties that are superior to those of shape-memory metal alloys, they have poor intrinsic mechanical properties limiting their possible applications in multiple scenarios. Therefore, to improve these mechanical properties while trying to preserve the elastic nature of the network, different methods have been extensively studied, such as higher crosslinking, changing deformation conditions, and adding fillers to the polymer matrix.

Reinforcing fillers can improve the mechanical behavior of the material with an evident enhancement of Young’s modulus, strength, and toughness [34]. This enhancement has also been successfully investigated in shape-memory polymers [35]. Moreover, for shape-memory composites, fillers can perform a secondary function as crosslinking agents. In this case, they not only reinforce but also improve shape-memory properties. Fillers that improve the thermal conductivity of the polymer matrix can have a positive effect on the thermal-activated shape-memory effect. Besides the reinforcement effect, the addition of electrically conductive fillers, such as carbon black [36], carbon nanotubes [37], and graphene [38,39,40], to the isolating polymeric matrix broadens the applicability of these materials, with the highlight of enabling or enhancing athermal stimuli-active effects, such as the electroactive shape-memory effect by Joule heating [41]. These composites have a wide variety of applications, such as antistatic materials, electromagnetic interference protection materials, sensors, actuators, and drivers [42]. Conductive elastomers have been the focus of considerable research over the past two decades due to foreseeable applications in technologies as diverse as biological and chemical sensors, antistatic coatings, and electromagnetic shielding [43,44].

Among the conductive fillers mentioned above, carbon black (CB) is one of the most commonly used conductive fillers due to its superior characteristics, such as its high electrical conductivity, chemical stability, and low cost [45]. However, normally, a very high content of CB (~15–30 wt%) is required to reach the conductive percolation threshold in a polymeric matrix, which can complicate its processing or cause a drop in its mechanical properties, further increasing its cost [36]. Additionally, the high content of reinforcing fillers sometimes leads to negative results in some vulcanization properties, such as compression set and the loss of hysteresis (or heat accumulation). Therefore, new reinforcing fillers with a relatively high specific surface area and/or aspect ratio have been introduced, for example, nanoclays and carbon nanotubes (CNTs) [46]. With these fillers, due to the high surface area of nanomaterials, the filler content required for any property can be markedly decreased, while good dynamic mechanical properties of the rubber can still be preserved. As such, a balance between processability and static and dynamic mechanical properties is possible. These polymer nanocomposites have become the preferred choice of polymer researchers for the design of lightweight compounds [47,48]. Moreover, the incorporation of nano-reinforcements significantly increases thermal stability [49]. CNTs, due to their exceptional mechanical properties and other functional properties of great interest, have become prime candidates for many novel applications [50,51,52]. Multiwalled carbon nanotubes (MWCNTs) have attracted more attention due to their low production cost and high performance. With the addition of MWCNTs, it has been possible to induce electrical conductivity in various elastomers, such as PU/PLA polymer blends [53], the styrene–butadiene–styrene (SBS) block copolymer [54], and polyethylene and EVA blends, with a percolation threshold of 3 wt% of MWCNTs [55]. Electroactive shape-memory materials provide the possibility of having an additional stimulus, as they are connected to a closed electrical circuit without the need to physically access the material, and the possibility of having precise control in the activation. All of this facilitates their use as sensors and actuators, where electricity is an easily implementable stimulus [56,57]. Percolation threshold for having conductivity in elastomeric samples needs higher MWCNT contents. Bernal-Ortega et al. achieved good electrical properties above 10 phr for natural rubber and styrene-butadiene rubber nanocomposites. However, MWCNT con-tents higher than 15 phr were an issue for the vulcanization process of the compounds, due to MWCNT aggregation and the high viscosity of the rubber matrix [58].

In recent years, the effect of the introduction of nano-fillers in XNBR matrices has been studied for different purposes [59,60]. Tian et al. [61] developed a material with high dielectric constant and low dielectric loss based on graphene oxide (GO) and XNBR. The dielectric constant of the composite started to increase with very low filler content (0.25 phr). Liu et al. [62] noncovalently modified graphene with sodium humate and used the modified graphene sheets on an XNBR matrix with the presence of MgO as an ionic crosslinker. Wang et al. [63] improved the thermal stability, conductivity, and thermal diffusivity of XNBR by incorporating functionalized graphene (1.6 phr) into an XNBR matrix. Preetha Nair et al. incorporated MWCNT in XNBR latex and obtained percolation behavior, and conductivity increased by about 10 orders of magnitude [64].

This research focuses on the following three main objectives: (i) to study the influence of adding conductive fillers (carbon black) and nano-fillers (multiwalled carbon nanotubes) on the network structure and shape-memory properties of XNBR compounds crosslinked with covalent and ionic bonds; (ii) to compare the reinforcement efficiency of the different fillers by investigating their viscoelastic behavior, mechanical properties, and electrical properties; and (iii) to develop a proof of concept of new advanced applications based on ionic elastomers with improved thermal and electrical conductivities.

## 2. Materials and Methods

### 2.1. Materials

Samples were prepared using XNBR Krynac^®^ X 740, (Lanxess Elastomers SAS, Stuttgart, Germany), with a proportion of 27 wt% of acrylonitrile and 7 wt% of carboxylic groups as a matrix. Furthermore, 4 parts per hundred of rubber (phr) of magnesium oxide, MgO Elastomag^®^ 170 Special (Akrochem, Akron, OH, USA), and 0.5 phr of dicumyl peroxide (DCP) (Aldrich, St. Louis, MO, USA) were used as crosslinking agents. In addition, all samples had 1 phr of stearic acid as a processing aid.

The reinforcing fillers used were conductive carbon black (CB) Black Pearls^®^ 2000 (Cabot, Boston, MA, USA), with high specific surface, and multiwalled carbon nanotubes (MWCNTs) NC7000 (Nanocyl^®^), with an average diameter of 9.5 nm and an average length of 1.5 µm. For MWCNT-reinforced compounds, filler content was limited to 15 phr due to the high viscosity reached during compounding.

The samples used are presented in Table 1.

### 2.2. Preparation of Rubber Compounds

All samples were prepared on a Gumix laboratory two-roll mill, with a cylinder diameter of 15 cm, a length of 30 cm, and a friction ratio of 1:1.15. The fillers were incorporated after rubber mastication and introduction of magnesium oxide and stearic acid. As such, DCP was left as the last ingredient to avoid possible pre-vulcanization since the samples were heated by friction. Later, all samples were vulcanized in a hydraulic press for 60 min at 160 °C.

### 2.3. Characterization Methods

The vulcanization process was studied using a rubber process analyzer, RPA 2000, from Alpha Technologies (Wiltshire, UK), where a deformation of 6.98% at a frequency of 1.667 Hz was applied for 2 h at 160 °C. Additionally, frequency-sweep experiments were carried out in the RPA after the vulcanization process. The shear modulus was measured at frequencies from 0.002 to 33.33 Hz at temperatures from 40 to 230 °C. All data were then combined in a master curve using the frequency−temperature superposition [65], obtaining the stress relaxation of the modulus in a wide range of frequencies.

Tensile test experiments were determined using a universal mechanical tester (Instron 3366 series, Norwood, MA, USA). Dumbbell geometry (type 4) test specimens were prepared as 1 mm thick, guided by ISO 37. The strain speed was 200 mm·min^−1^. For each compound, five specimens were tested at room temperature.

Dynamic mechanical measurements and bulk shape-memory effect of vulcanized samples were carried out in a TA Q800 dynamic mechanical analyzer (TA Instruments, Inc., New Castle, DE, USA). Dumbbell geometry (type 4) samples guided by ISO 37 with 1 mm thickness were used for both experiments. Dynamic mechanical experiments were loaded under tension with an oscillatory deformation. Temperature was registered from −70 to 200 °C, with a heating rate of 2 °C·min^−1^. For the study, an amplitude of 15 μm, frequency of 1 Hz, and “force track” (the ratio of static to dynamic forces) of 108% were applied.

Shape-memory behavior was quantitatively characterized using a thermo-mechanical cycling method divided into five steps, as widely explained in previous studies [17], controlling the applied force under tension. To quantify the shape-memory behavior, fixing ratio (*R_f_*) and recovery ratio (*R_r_*) were calculated according to [66]:(1)$$R_{f}\left(N\right)=\frac{\varepsilon_{u}\left(N\right)}{\varepsilon_{m}\left(N\right)}\times 100\%.$$
(2)$$R_{r}\left(N\right)=\frac{\varepsilon_{u}\left(N\right)-\varepsilon_{p}\left(N\right)}{\varepsilon_{u}\left(N\right)-\varepsilon_{p}\left(N-1\right)}\times 100\%$$
where *ε_m_* represents the strain before unloading, *ε_u_* is the strain after unloading, *ε_p_* represents the permanent strain after heat-induced recovery (including an isothermal of 30 min), and *N* corresponds to the cycle number of the test. A value of 100% indicates complete strain fixing/recovery of the sample. For cycle 1, *ε_p_*(0) takes the value after the initial isothermal.

Moreover, electrically activated shape memory was characterized with a bending test. The samples used to measure this application of the nanocomposites were “U”-shaped sheets. Temporary shape was fixed by heating the material in an oven for 5 min in a mechanized steel mold with 2 cm-radius curvature as found in [53]. During these tests, the main parameter is the angle of the SMP as it is bent in its temporary shape after waiting for 30 min free of strain. The value of the degree of shape recovery (*R_r_*) is calculated as
(3)$$R_{r}\left(N\right)=\frac{\theta_{0}-\theta_{N}}{\theta_{0}}\times 100\%$$
where *θ*_0_ is the temporarily fixed angle, and *θ_N_* is the residual angle obtained during the cycle recovery process.

Electrical conductivity of the compounds was determined using an ALPHA high-resolution dielectric analyzer (Novocontrol Technologies GmbH, Hundsangen, Germany). A frequency range window of 10^−1^–10^7^ Hz at room temperature was used. The measurements were performed on samples of 2.5 × 2.5 × 0.1 mm^3^ between two parallel gold-plated electrodes.

Thermal conductivity was measured under stationary conditions using a heat flow meter, model FOX 50 (Lasercomp—TA Instruments, Wakefield, MA, USA), according to ASTM C518 and ISO 8301 standards. The measurements were performed on cylindrical specimens of 45 ± 0.1 mm diameter and 8 ± 0.1 mm thickness. At least three measurements were carried out for each sample, with an experimental error of less than 2% of the absolute value.

## 3. Results and Discussion

### 3.1. Formation of Rubber Networks

The formation of the ionic and covalent crosslinking networks during the vulcanization process was studied through the monitoring of torque evolution with time at a temperature of 160 °C in a rheometer, and it is reflected in Figure 1.

The vulcanization curves show a fast increase in the elastic component of the torque (*S*’) without any visible scorch time because of the fast formation of the ionic and covalent crosslinks at the selected vulcanization temperature. After that, a marching plateau is observed for all samples, demonstrating that the addition of fillers to the elastomeric matrix does not limit the process of ion-pair formation and reorganization over time and maintains the predominant ionic character studied in a previous work [67]. Nevertheless, the maximum value of *S*’ increases with the added fraction of filler according to the reinforcing effect of both CB and CNT. The correct incorporation of the fillers to the matrix was studied by thermogravimetric analysis (Appendix A). The images of the distribution and size of the fillers obtained using FEG-SEM can be found in Appendix A.

To observe the reaction of the carboxylic groups with MgO, FTIR-ATR was used. As studied in previous works, the formation of covalent crosslinks (by adding DCP to the system) does not influence the IR band of the carboxylic groups of the rubber matrix [32,68]. Therefore, it can be used to study the reaction of these moieties with metal oxide. The formation of ionic bonds in the elastomeric matrix, produced by the chemical reaction between MgO and the carboxylic groups of XNBR rubber, can be observed in the range between 1800 and 1500 cm^−1^ (Figure 2). The bands corresponding to the carboxylic acids of the matrix (1730 and 1697 cm^−1^) disappear, regardless of the nature and content of the filler introduced into the material, and a new band appears at a vibration frequency of 1580 cm^−1^, corresponding to the magnesium carboxylate salts generated.

The resolution of the signals fades with an increase in CB content. The normalized signals are not as clear for the highest CB filler contents (25CB and 30CB).

The satisfactory reaction of the carboxylic groups of the matrix with magnesium oxide in the presence of the different fillers is a sign of the existence of the phase separation phenomenon produced by ionic moieties and, as such, managing to maintain the ionic transition necessary to obtain a shape-memory effect in these materials [67]. The mechano-dynamic behavior of the materials was characterized at a frequency of 1 Hz, performing a temperature sweep between 70 and 180 °C to define the polymer transitions (Figure 3).

Through the variation in tan δ with temperature (Figure 3a), a thermal transition at a temperature around 5 °C, attributed to the glass transition of the polymer, is clearly distinguished in all samples. This temperature increases by a maximum of 2–3 °C when introducing the CNT fillers, while for carbon black, this variation is only around 1 °C. The intensity of this maximum peak and the appearance of the second transition, assigned to ionic transition, are progressively lost with the filler content in the composite material. Despite this, the behavior of the samples with temperature is similar in all the materials. This effect is better appreciated when observing the evolution of the storage modulus (*E*’) with temperature in Figure 3b. The introduction of reinforcing fillers increases the rigidity and, hence, the *E*’ values of the material proportionally to their content, with this effect being more visible when exceeding the glass transition temperature. However, the changes in the slope remain practically unchanged; both “pseudo-plateaus” are observed both when exceeding the *Tg* and at the end of the broad ionic transition (see Appendix A for *T_g_ T_ionic_* and *E*’ values below and above *T_g_*).

The rheometric measurements were used to characterize the dynamics of the networks with temperature by using the time–temperature superposition principle [50] to create master curves representing the variation in elastic modulus (G’) in a wide range of frequencies (Figure 4).

As previously mentioned, ionomers exhibit viscoelastic properties similar to those of molten polymers with high molecular weight and highly entangled chains or in concentrated solutions [69,70]. However, ionic elastomers flow through the “ion-hopping” mechanism without breaking all their ionic associations simultaneously [71,72,73]. Chain creep is a slow process (visible at low frequencies) that requires high temperatures, since the ion-hopping mechanism needs to be fast enough for the elastomeric chains to relax.

Assuming that the residence time of the ion pairs in each ionic aggregate is the same regardless of its structure (ion pairs, multiplets, or ionic clusters), the chain dynamics slows down with the addition of dispersed fillers in the matrix (i.e., terminal relaxation time increases) due to restrictions to movement or steric impediments in the mobility of ionic interactions (Figure 4). This impairs the reprocessability of the materials and is, in turn, a clear indicator of the reinforcing effect of the introduction of these fillers.

### 3.2. Physical Properties

The effect of fillers on dynamics is directly related to the physical properties of composites at the macroscopic level. As studied in previous studies, XNBR ionic elastomers have excellent mechanical properties at room temperature since the association between MgO and the carboxylic groups of the matrix acts as physical crosslinks while also making the effect of reinforcing fillers [68].

For this reason, it is obvious that the introduction of external reinforcing fillers produces an increase in the modulus of the material, although, in this case, it will gradually limit the deformation capacity of the material. Figure 5 shows the stress–strain curves for all the materials with the different fillers in which their reinforcing effect is appreciated, with a progressive increase in the modules of the material, as well as a reduction in the maximum deformation with an increase in filler content.

If the evolution of the modulus and the tensile strength of the material is observed in detail, certain differences can be distinguished between the reinforcing fillers, showing that CNT has a higher reinforcing effect than that of CB. In both cases, the incorporation of reinforcing fillers translates into an improvement in the tensile strength of the materials, but it has a detrimental effect on their elastic properties, limiting the maximum strain and, hence, the ultimate tensile strength (see Table 2). Despite this, deformations greater than 300% can be achieved in all cases, valid for many applications, in our case, for their application as elastomeric materials with shape memory.

### 3.3. Shape-Memory Properties of Elastomeric Composites

In this part of the study, we investigate the influence of the introduction of reinforcing fillers in the XNBR matrix crosslinked with ionic associations between MgO salt and the carboxylic groups of the matrix, responsible for the fixation of the temporary shape, and the covalent unions produced by the introduction of DCP, responsible for the recovery of the original shape.

#### 3.3.1. Effect of the Incorporation of Reinforcing Fillers on the Shape-Memory Properties of Elastomers

The ability of ionic interactions to act as temporary physical crosslinks depends on temperature, time, and applied stress. However, the covalent bonds present thermostable behavior and tend to recover the original and permanent shape of the samples. These mechanisms are responsible for the activation of the shape-memory properties in the materials, where heating the material above the ionic transition temperature can cause the ionic interactions to become ineffective, and the shape of the material can be modified into a new shape. With the new shape, if the material is cooled down, those same ionic interactions become effective again and retain the temporary shape until the material is again heated above the ionic transition temperature. Both processes can be altered with the introduction of dispersed reinforcing elements in the matrix, which will hamper the elastic characteristics of the network and, in turn, reduce the exploration space accessible to chains and ionic pairs or aggregates of a dynamic nature.

According to the experimental procedure described in [32], shape-memory tests with four memory cycles were carried out for each sample with an increase of 50% in strain for the temporary shape. Figure 6 shows the complete characterization of one of the samples (containing 15 phr CNT). In our previous work, up to 10 cycles were carried out to demonstrate stability with these SM cycles [32].

The fixation of the temporary shape (*R_f_*) and the recovery of the permanent shape (*R_r_*) values were obtained with Equations (2) and (3), and those values are presented in Figure 7 for all the composites tested.

As can be seen, the addition of reinforcing fillers improves the fixing properties of the temporary shape, reducing the negative effect produced by the viscoelasticity of the pristine sample. Therefore, the incorporation of a higher filler content has a direct effect on fixation. Comparatively, the introduction of carbon nanotubes results in the best fixing properties being obtained (between 4 and 10% improvement with respect to the unfilled material for the CNT contents studied). This fact may be due to the good dispersion and/or the alignment of the nanotubes in the direction of the deformation, making it difficult to lose this new transitory shape. The 15CNT sample achieved fixing ratio values above 90% in every cycle. All data values can be seen in Appendix A.

Furthermore, contrary to what occurred when the content of the crosslinking agents varied, where the improvement of one of the shape-memory properties had a negative effect on the other as previously reported [32], the addition of fillers also translated into a general improvement in the recovery of the permanent shape. Although this capacity is impaired with the addition of large contents, the introduction of fillers produces, in all cases, proportions above 25 phr of CB, an improvement in recovery with respect to its unfilled counterpart of up to 10% in the first test cycle.

This conclusion is even clearer in Figure 8, where the relationship between the fixing and recovery capacities of the nanocomposites is shown in the same graph.

Summarizing the results shown in Figure 8 for the different shape-memory elastomers reinforced with carbonaceous fillers, the best balance of properties belongs to the nanocomposites with 10 or 15 phr of CNTs, whose fixation exceeds 90% and recovery capacity exceeds 86% for the first cycle, with these values progressively increasing with the number of cycles. Reinforcement with nanotubes is the most effective approach for this application, while reinforcement with carbon black has a similar influence but requires a filler amount of more than double that of the nanotubes (20–25 phr), which has repercussion on other properties (e.g., processing properties and elongation at break).

#### 3.3.2. Effect of the Incorporation of Fillers on the Thermal Conductivity of Elastomeric Nanocomposites

Figure 9 shows the experimental values of thermal conductivity. All samples show a linear dependence on temperature in the analyzed range between 20 and 80 °C. The thermal conductivity obtained increases with the addition of an increasing content of charge in its structure. This effect is higher when carbon nanotubes are introduced with respect to carbon black filler. With them, an improvement in this property of more than 25% is achieved with respect to the unfilled material.

The slight improvement in the thermal conductivity of the samples should contribute to the shape-memory properties of the material, since the stimulus used to activate this effect is a thermal one. To analyze this improvement, the instantaneous recovery ratio, *iR_r_*, and the instantaneous recovery speed, *V_r_*, were calculated as a function of time following Equation (2) and
(4)$$V_{r}\left({\%\cdot \min}^{-1}\right)=f\left(T\right)=\left(-\frac{\partial \varepsilon_{p}\left(T\right)}{\partial T}\right)\cdot (10\;{}^{\circ}C\cdot \min^{-1})\times 100\%$$
where $\frac{\partial \varepsilon_{p}\left(T\right)}{\partial T}$ is the temperature derivative of the strain from the strain recovery data, and 10 °C·min^−1^ is the heating rate. The temperature derivative of the strain is negative due to the decreasing strain recovery curve, so the minus sign is used to convert the instantaneous velocity into a positive value. Both variables (*iR_r_* and *V_r_*) depend on the recovery temperature, *T*, in each moment, and they are represented in Figure 10 for samples reinforced with CNTs, as they presented the best thermal conductivity values.

Slight improvements can be observed, both a faster response speed and a maximum speed, which appear at times (and, therefore, at temperatures) slightly lower than when CNT is added as a filler. Even so, the limitation of the ramp in controlling the temperature in the equipment (10 °C·min^−1^) influences these results; as can be seen in Figure 10a, the evolution of instant shape recovery is linked in a certain way to the evolution of the chamber (running) temperature, so big differences cannot be appreciated. Instant heating to 150 °C should be ideal to unveil these uncertainties.

#### 3.3.3. Electroactivation of the Shape-Memory Effect in Elastomeric Nanocomposites

Most unfilled SMPs are inert to electric current, thus preventing the shape-memory effect from being directly electrically induced through resistive heating. Furthermore, when introducing fillers, it is generally difficult to control the dispersion of the nanoparticles to achieve a continuous and regular conductive path or network in the polymer matrix, resulting in high electrical resistivity.

The electrical conductivity of the elastomeric nanocomposites varies with the concentration of the introduced nano-fillers as presented in Figure 11.

As the filler content increased, the average electrical conductivity of the nanocomposite increased from ~10^−11^ to ~10^−4^ S·cm^−1^. This range is sufficient to achieve the heating of the materials by the Joule effect, as has been previously demonstrated [74].

The electrical conductivity in the rubber compound increases with a higher CNT or CB fraction, because more conductive paths for electrical currents are formed in the rubber matrix. Therefore, both the current transport capacity and the amplitude of the electric current increase. Furthermore, by increasing the number of possible conductive paths through the fillers, the probability of forming shorter distances for electrons in the electrical current increases, resulting in increased electrical conductivity.

As shown, the electrically conductive properties of SMP nanocomposites improve with the addition of CNT fillers. Therefore, an electrical actuation and an electrically activated shape-memory effect could be achieved in these nanocomposites through thermal activation by the Joule effect [75].

For the practical application of SMPs, their shape recovery performance is extremely important and is generally evaluated using a bending test. The effect of the electrical drive and recovery of the original shape was investigated in the nanocomposite with 15 phr of CNTs since it presented the best electrical conductivity data. “U”-shaped sheets were fabricated to measure this application for different samples. During these tests, the main parameter is the angle of the SMP as it is bent in its temporary shape [76]. Due to the elastomeric characteristics of the nanocomposites, the samples were deformed under flexion under an external force by using a steel mold, and they were subsequently heated in that position to 150 °C to temporarily fix their shapes, maintaining the deformation and the temperature for 15 min.

After cooling to room temperature, the deformation was released, and the temporary shape was fixed. The recovery of the electric-field-activated shape was observed by recording images with a video camera while a constant voltage of 50 V (DC) was applied to the tested sample. Figure 12 shows the sequential images of the electroactive behavior during the process of recovery of the shape of the tested nanocomposite (15 phr of CNTs) (see the complete video in the Appendix A).

As presented in Figure 13, the temperature that the sample reached was measured using a thermocouple in contact with the surface of the specimen, and the recovery of the instantaneous shape was calculated as a function of this temperature over time through Equation (2), with the application of a current of 50 V.

The original shape of the sample was almost completely recovered in ~15 min when a 50 V electric field was applied. The process was previously tested with lower electric fields, but the sample was not heated enough to recover its original shape. The shape recovery speed therefore strongly depends on the magnitude of the applied voltage, provided that the current intensity remains constant. As such, the nanocomposites could be heated more quickly or to a higher temperature with application of a higher voltage. However, it was difficult to control the temperature of the sample throughout the process for other electric fields.

The electrical actuation efficiency, which is a measurement of the degree of conversion of the transformation of electrical energy into resistive heat energy, was not systematically investigated. Resistive heat is not completely transferred to the SMP matrix due to the practical dissipation of energy in air. Therefore, high actuation efficiency is essential for the sustainable development and practical application of SMPs powered by electricity. Hence, the determination of shape recovery behavior could not be fully characterized. However, the possibility of using an electric field to activate the recovery of the original shape in this type of elastomeric nanocomposites was demonstrated.

## 4. Conclusions

Shape-memory elastomeric composites, based on an XNBR matrix with ionic and covalent crosslinks and the introduction of carbon black and multiwalled carbon nanotubes as reinforcing fillers, were obtained.

The incorporation of reinforcing fillers helped to improve the shape-memory behavior of these ionic elastomers, both in the fixation of the temporary shape and in the recovery of the original shape, over progressive cycles. The introduction of carbon nanotubes resulted in the best fixing properties being obtained (reaching 10% improvement compared to the unfilled material in both properties), achieving fixing and recovery ratios above 90%. Moreover, the elastomeric nanocomposites showed an improvement in thermal conductivity. This affects the response speed of the material when recovering the original shape, so better control of the response to the thermal stimulus is achieved. This fact opens the possibility of studying the temperature memory effect of these compounds, as the materials react at temperatures close to their deformation temperature.

Finally, the introduction of electrically conductive fillers, e.g., carbon nanotubes and carbon black, provides the possibility to develop ionic elastomers with shape-memory properties that can be activated by an electric current, based on the heating of the material by the Joule effect, achieving a fast and clean stimulus requiring only a current source. This method opens up the field for the recovery of the selective shape, in specific areas, by specific heating between the entry and exit points of the current. As such, we introduced a path to be explored in depth in future studies.

## Figures and Tables

**Figure 1 polymers-14-01230-f001:**
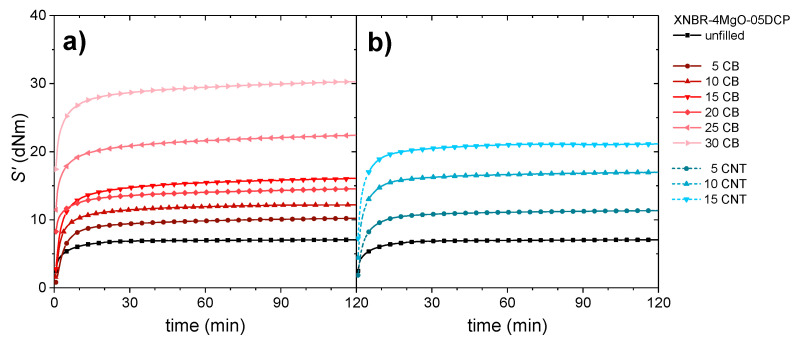
Evolution of the vulcanization curve (through the elastic component of the torque, *S*’) of the systems reinforced with different amounts (phr) of (**a**) CB and (**b**) CNT.

**Figure 2 polymers-14-01230-f002:**
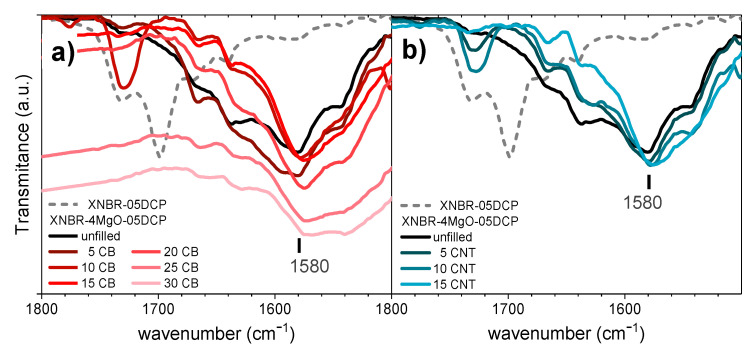
Details (1800–1500 cm^−1^) of the infrared spectrum (FTIR-ATR) of the XNBR samples crosslinked with only 0.5 phr of DCP (dashed line), with 4 phr of MgO, and with the incorporation of increasing amounts of (**a**) CB and (**b**) CNT reinforcing fillers.

**Figure 3 polymers-14-01230-f003:**
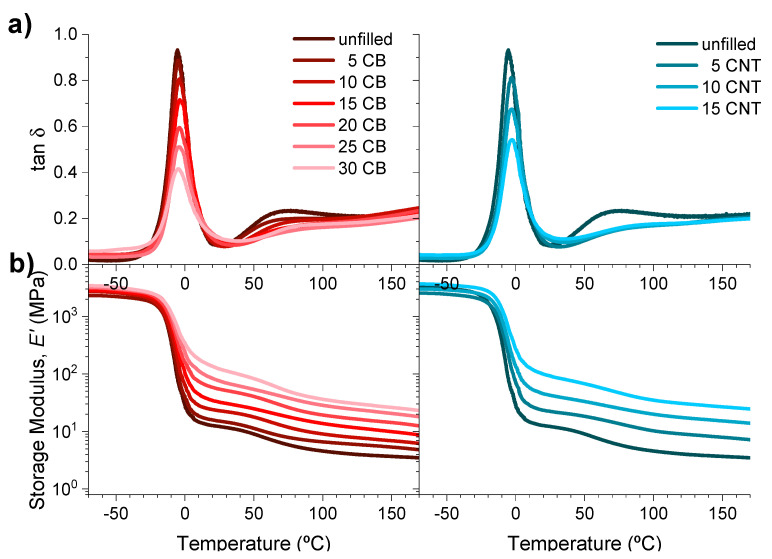
(**a**) Variation in the loss tangent (tan δ) and (**b**) of the storage modulus, with the temperature at a frequency of 1 Hz, for the XNBR-4MgO-0.5DCP samples with the introduction of increasing amounts of CB and CNT reinforcing fillers.

**Figure 4 polymers-14-01230-f004:**
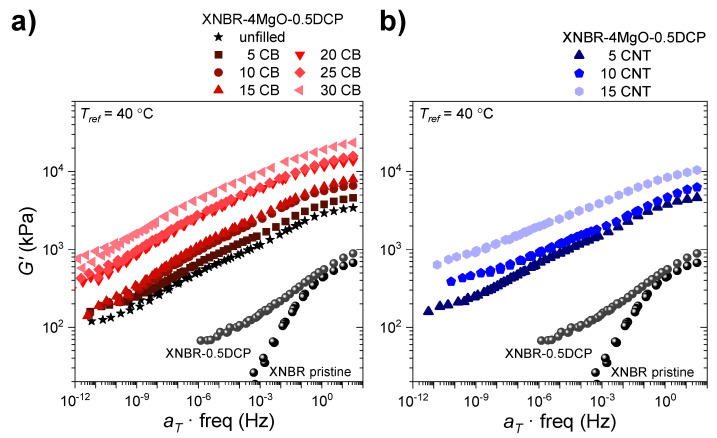
Master curves in which the elastic modulus (*G*’) is represented with respect to the frequency of the XNBR-4MgO-0.5DCP samples with (**a**) CB contents between 0 and 30 phr and (**b**) CNT contents between 0 and 15 phr. The curves were obtained from the principle of time–temperature superposition with *T_ref_* = 40 °C.

**Figure 5 polymers-14-01230-f005:**
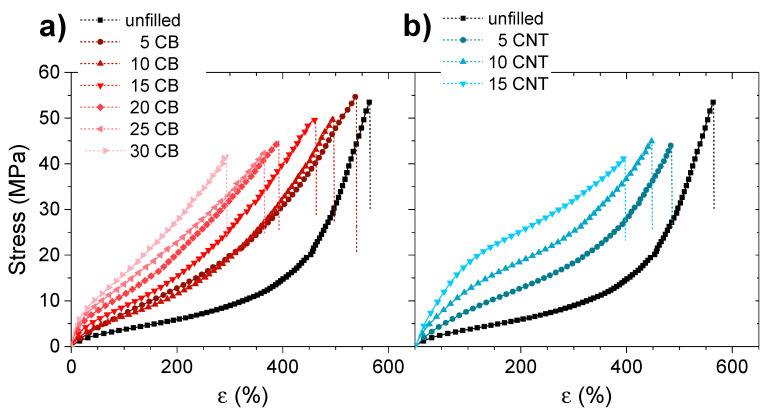
Stress–strain curves obtained for the XNBR-4MgO-0.5DCP samples with the introduction of increasing contents of (**a**) CB and (**b**) CNT.

**Figure 6 polymers-14-01230-f006:**
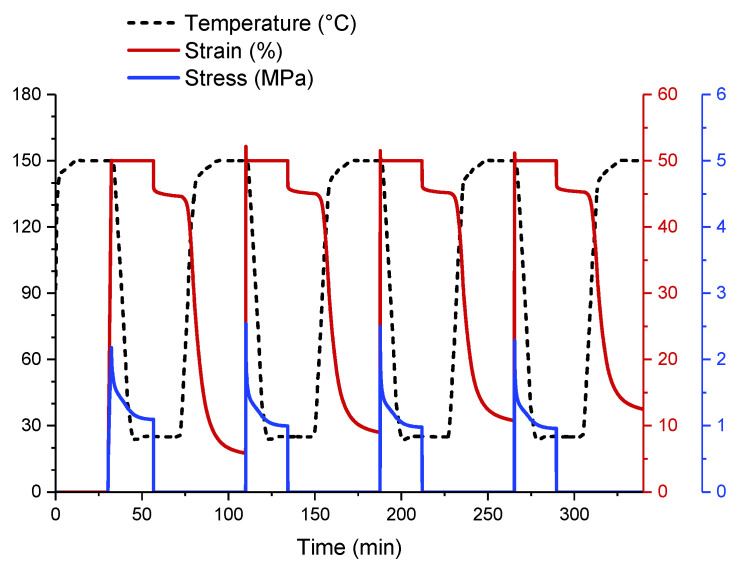
Shape-memory test carried out on sample 15CNT. Stress, strain, and temperature are registered simultaneously.

**Figure 7 polymers-14-01230-f007:**
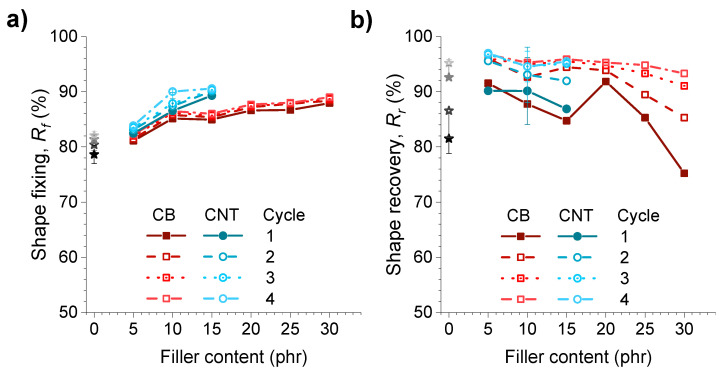
(**a**) Degree of fixation of the temporary shape and (**b**) degree of recovery of the permanent shape for the sample XNBR-4MgO-0.5DCP (unfilled), and with the incorporation of increasing content of CB and MWCNT for 4 consecutive cycles.

**Figure 8 polymers-14-01230-f008:**
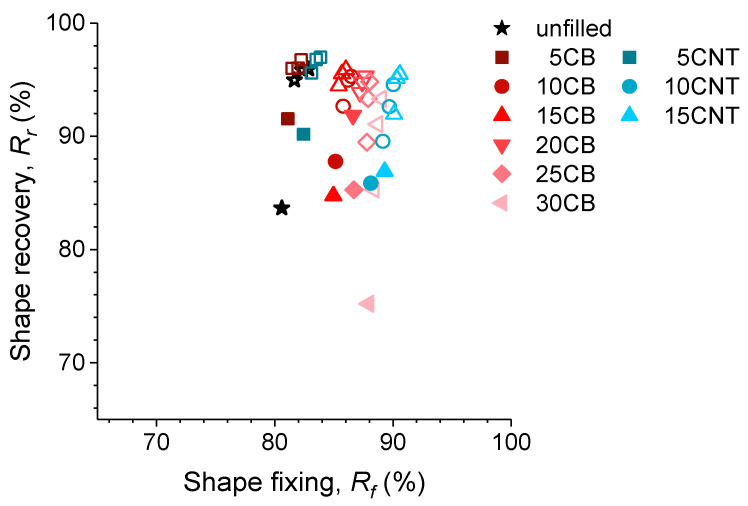
Relationship between fixation and shape recovery values for the different nanocomposites studied. Solid symbols mark the data for the first cycle, while hollow symbols represent the following three cycles.

**Figure 9 polymers-14-01230-f009:**
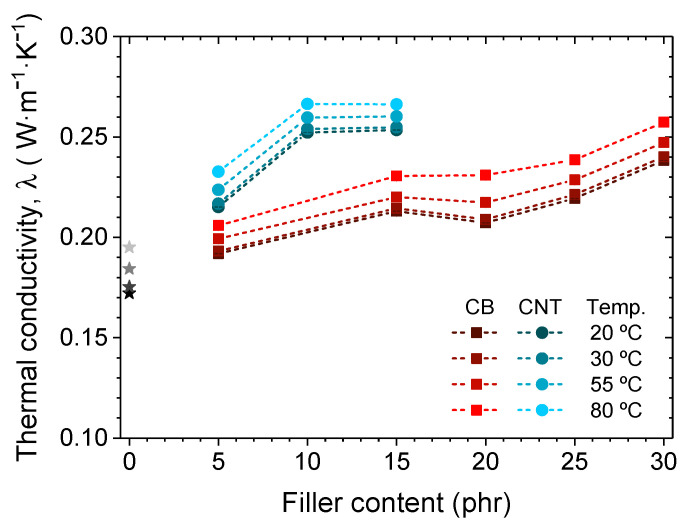
Thermal conductivity of the pristine XNBR-4MgO-0.5DCP, and with an increase in content of CB and CNT fillers, measured at 4 reference temperatures.

**Figure 10 polymers-14-01230-f010:**
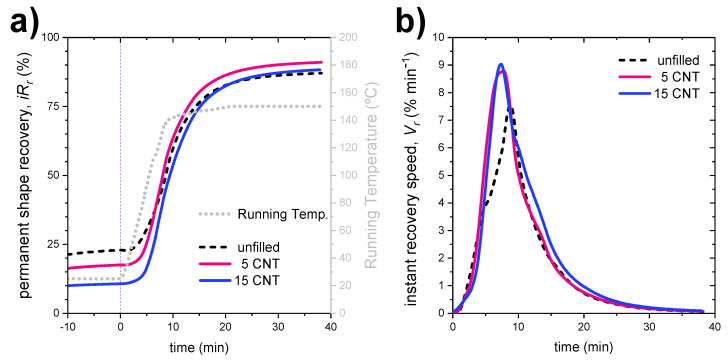
(**a**) Instantaneous recovery ratio curves as function of recovery temperature and (**b**) instantaneous recovery speeds as function of recovery temperature for samples reinforced with increasing amounts of CNTs.

**Figure 11 polymers-14-01230-f011:**
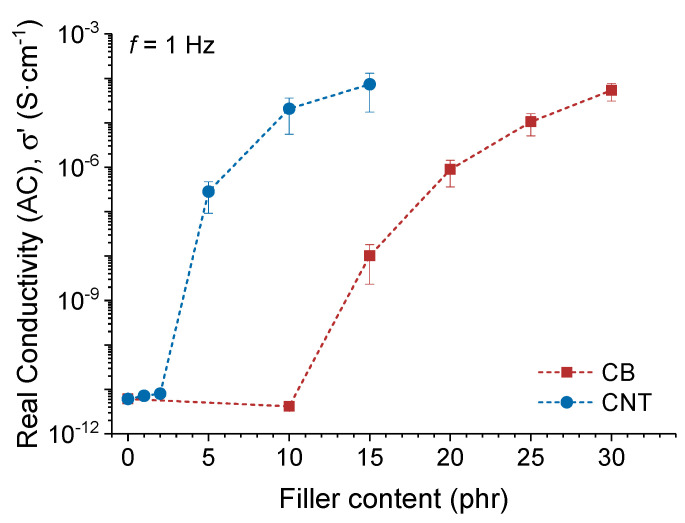
Real electrical conductivity (σ’) in alternating current (AC) at a frequency of 1 Hz for pristine XNBR-4MgO-0.5DCP and with increasing filler content of CB and CNT. The error bar represents a standard deviation, illustrating the variability of the tested results.

**Figure 12 polymers-14-01230-f012:**
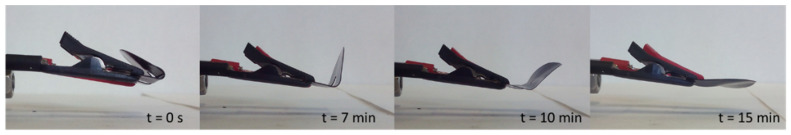
Electroactive shape recovery behavior of the XNBR nanocomposite with 15 phr of CNTs as a function of the time of application of a constant voltage of 50 V.

**Figure 13 polymers-14-01230-f013:**
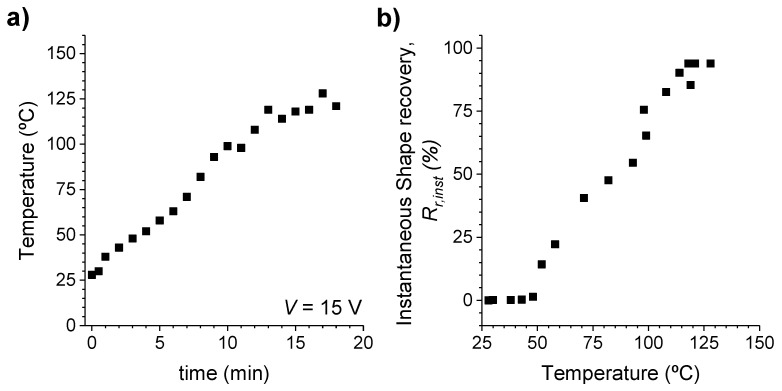
(**a**) Surface temperature of the XNBR sample with 15 phr CNTs when a current of 50 V is circulated through it. (**b**) Instantaneous recovery of the shape as a function of the temperature reached by the sample.

**Table 1 polymers-14-01230-t001:** Formulations of the different networks studied. Quantities are expressed in parts per hundred of rubber (phr).

Sample	XNBR	MgO	DCP	Stearic Acid	Filler	
	phr	phr	phr	phr	phr	Type
XNBR-4MgO-0.5DCP (unfilled)	100	4	0.5	1		
5CB	100	4	0.5	1	5	CB
10CB	100	4	0.5	1	10	CB
15CB	100	4	0.5	1	15	CB
20CB	100	4	0.5	1	20	CB
25CB	100	4	0.5	1	25	CB
30CB	100	4	0.5	1	30	CB
5CNT	100	4	0.5	1	5	CNT
10CNT	100	4	0.5	1	10	CNT
15CNT	100	4	0.5	1	15	CNT

**Table 2 polymers-14-01230-t002:** Tensile strength and elongation at break values obtained from stress–strain curves of the samples.

Sample	Tensile Strength		Elongation at Break	
	MPa		%	
Unfilled	51	±3	560	±10
5CB	52	±5	506	±37
10CB	49	±6	493	±10
15CB	50	±2	462	±17
20CB	45	±2	399	±35
25CB	44	±1	353	±39
30CB	42	±2	308	±23
5CNT	45	±3	493	±34
10CNT	44	±4	453	±27
15CNT	42	±2	389	±18

## Data Availability

Not applicable.

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
