# Peer review of "Shape-Memory Composites Based on Ionic Elastomers"

_polymers, 2022, doi:10.3390/polym14061230_

Round 1

Reviewer 1 Report

(Major revision) This manuscript describes the shape memory nanocomposites based on ionic elastomers. The manuscript is well designed. However, many things need to revise before the next level of publication.

Authors claimed that it is nanocomposites. but there is no characterization of nanocomposites so authors need to incorporate the FESEM or TEM results.

The abstract should be rewritten in a quantitative manner.

All figures' clarity should be enhanced.

Line 25-29, insert more references to support each SMP. Water Induced SMP is missing. Insert this citation for water-induced SMP Carbohydrate Polymers 257 (2021) 117633; Composites Science and Technology, 2022, 109255; European Polymer Journal, 2021, 161, 110823

Why Electric SMPs are more important than other SMPs, please discuss it in the introduction section.

Line 32 SMP should be SMPs.

Section 2, please make it section-wise like Materials, methods, and so on.

How do authors make U shape? Before SMP test how about stability for U shape in the air? This information is included clearly in the manuscript.

Line 51-71, 155-181, 188-197, and 433-455 authors directly copied each wording from other publications. Kindly replace it in your words.

Table 1 Sample representation is not clear. Please arrange it.

For nanocomposite, authors need to add TGA data.

In table 1 author presented CB 5-30 phr but why did authors only prepare 5-15 in the cases of CNT.

Line 164, 20 mm/min should be changed into mm min-1. These types of errors authors should correct throughout the manuscript.

Crystallinity data must be given by authors either from DSC or XRD (XRD is preferable) accordingly Journal of Luminescence, 228 (2020) 117593.

Line 231-235, write in a better way.

Which sample has a better SMP? Authors should have presented all sample's time recovery time? What about stability? How many cycles it can perform without the loss of stability.

Why references are errors. It is difficult to comment.

In Fig. 1, why 15CB (3rd curve from bottom) have stopped around at 90 mins.

Figure 2 caption needs to rewrite. ‘Increasing proportions’.? Check other figure captions.

Fig. 2 y-axis it is transmittance % or a.u. if a.u., no need to mention 0-100 as tick levels.

From Fig. 3 authors should estimate the Tg and storage modulus and Tan delta in the tabular format.

All subfigures must be separated as Fig. 'x' a and b. Check other figures. Avoid symbol in the x-axis. please write as wavenumber.

The mechanism of SMP is missing!

Section 3.2, Young’s modulus EB and Tensile strength data must be provided by authors in tabular format.

Fig. 7 shapes recovery data and fixes data better to provide in tabular form. Fig. 8 also.

Authors need to provide a comparison table of Electroactive SMP related to present study materials with recent publications (Preferable within 5 years’ paper).

Cite recent paper. 2020, 2021, 2022

The conclusions section has so many paragraphs. Make it a maximum of two. Rewrite it precisely.

Check the reference style as per the Polymers journal guidelines. Abbreviation of Journal name should be in references part.

Author Response

Reviewer 1

(Major revision) This manuscript describes the shape memory nanocomposites based on ionic elastomers. The manuscript is well designed. However, many things need to revise before the next level of publication.

Authors claimed that it is nanocomposites. but there is no characterization of nanocomposites so authors need to incorporate the FESEM or TEM results.

FEG-SEM images were incorporated in supporting information document to visualize particles size, where MWCNT dimensions are in scale of nanometers. Nonetheless, the nano-size of filler particles in the rubber matrix is not actually a required condition for the main aim of the work. In addition, as it is well described in the literature, carbon black aggregates and agglomerates in rubber compounds does not achieve the nanoscale. For these statements, the term “nanocomposite” has been replaced along the document.

The abstract should be rewritten in a quantitative manner.

Abstract section was modified following reviewer 1 and reviewer 2 recommendations.

All figures' clarity should be enhanced.

Minor modifications were made trying to enhance figures’ clarity

Line 25-29, insert more references to support each SMP. Water Induced SMP is missing. Insert this citation for water-induced SMP Carbohydrate Polymers 257 (2021) 117633; Composites Science and Technology, 2022, 109255; European Polymer Journal, 2021, 161, 110823

More references were added in the introduction section with examples of different shape memory stimuli.

Why Electric SMPs are more important than other SMPs, please discuss it in the introduction section.

A paragraph about this topic was added in the introduction section

Line 32 SMP should be SMPs.

The abbreviation was corrected.

Section 2, please make it section-wise like Materials, methods, and so on.

Section 2 was modified following reviewer 1 recommendation.

How do authors make U shape? Before SMP test how about stability for U shape in the air? This information is included clearly in the manuscript.

A sentence was included in section 2 to clarify this point.

Line 51-71, 155-181, 188-197, and 433-455 authors directly copied each wording from other publications. Kindly replace it in your words.

Some modifications were made in the text as recommended by reviewer 1.

Table 1 Sample representation is not clear. Please arrange it.

Table 1 was modified for better clarity

For nanocomposite, authors need to add TGA data.

TGA data was added in supporting information document.

In table 1 author presented CB 5-30 phr but why did authors only prepare 5-15 in the cases of CNT.

Processability of rubber compounds with MWCNT contents higher than 15 phr is an issue due to MWCNT aggregation and the high viscosity of the rubber matrix. On the other hand, the electrical conductivity of rubber compounds with 15 phr of MWCNT where similar to those obtained with higher amounts of CB.

Line 164, 20 mm/min should be changed into mm min-1. These types of errors authors should correct throughout the manuscript.

Scientific notation was used all along the manuscript.

Crystallinity data must be given by authors either from DSC or XRD (XRD is preferable) accordingly Journal of Luminescence, 228 (2020) 117593.

This material is not semi-crystalline polymer, it is completely amorphous rubber and therefore it maintains its rubbery state throughout the entire temperature range (above and below the Ti). The only element that can present a certain order is the hierarchical structure of the ionic interactions, and this fact was deeply analysed through XRD experiments in previous work (Macromolecules, 2014, 47(16), pp. 5655–5667)

Line 231-235, write in a better way.

This paragraph was rewritten for better understanding.

Which sample has a better SMP? Authors should have presented all sample's time recovery time? What about stability? How many cycles it can perform without the loss of stability.

Some changes in section 3.3.1. were included in order to clarify this point.

Why references are errors. It is difficult to comment.

It seems that there was an error with final docx file uploaded to the platform to review. This problem has been fixed. We apologize for the inconvenience.

In Fig. 1, why 15CB (3rd curve from bottom) have stopped around at 90 mins.

Figure 1 was corrected with complete data

Figure 2 caption needs to rewrite. ‘Increasing proportions’.? Check other figure captions.

Figure captions were corrected along the manuscript.

Fig. 2 y-axis it is transmittance % or a.u. if a.u., no need to mention 0-100 as tick levels.

The y-axis of Figure 2 was corrected by removing tick levels

From Fig. 3 authors should estimate the Tg and storage modulus and Tan delta in the tabular format.

Table S1 contains data for Tg, Tionic and E’ values below and above Tg. This table was added to the supporting information section and it referenced in the manuscript.

All subfigures must be separated as Fig. 'x' a and b. Check other figures. Avoid symbol in the x-axis. please write as wavenumber.

These recommendations were considered in the revised manuscript.

The mechanism of SMP is missing!

A brief paragraph explaining SM mechanism has been added, and references to previous work explaining it in detail appear in the revised manuscript.

Section 3.2, Young’s modulus EB and Tensile strength data must be provided by authors in tabular format.

Tensile strength and EB data were incorporated as a new Table in the manuscript. Young's modulus is not reported in elastomers because it is not a significant parameter in rubber compounds.

Fig. 7 shapes recovery data and fixes data better to provide in tabular form. Fig. 8 also.

Shape memory data values were incorporated in supporting information section.

Authors need to provide a comparison table of Electroactive SMP related to present study materials with recent publications (Preferable within 5 years’ paper).

Shape memory elastomers using the ionic temperature as switching transition are rubbery in all the temperature range and they show a quite different properties when they are compared with other shape memory polymers where switching transition is based on Tg and/or Tm transitions. For that reason, a quantitative comparison between SM elastomers, SM thermoplastics or SM thermoset has not sense in this work, independently of the applied stimulus. The main idea of incorporating conductive fillers were i) improve the shape memory properties of these elastomers (compared to the unfilled compounds) and ii) demonstrate that electric stimulus can be incorporated in SM elastomers, as a proof of concept for further studies.

Cite recent paper. 2020, 2021, 2022

New references were added along the document with recent publications in the field.

The conclusions section has so many paragraphs. Make it a maximum of two. Rewrite it precisely.

Conclusion section was rewritten following reviewer 1 and reviewer 2 recommendations.

Check the reference style as per the Polymers journal guidelines. Abbreviation of Journal name should be in references part.

Citation style was automatically created by Mendeley add-on in Microsoft word.

Reviewer 2 Report

  1. Revise the abstract, no quantitative information is available.
  2. There is a manifest demand for soft actuators,[3,4] so, during last years, different 36 approaches have been made to get polymers which preserve an elastic nature in the 37 temporary shape.[5–7]. Discuss each cited reference individually.
  3. There is some error in the manuscript "Error! Reference source not found..'
  4. Revise the conclusion section.
  5. Compare the findings of current research work with past researc

Author Response

  1. Revise the abstract, no quantitative information is available.

Abstract section was modified following reviewer 1 and reviewer 2 recommendation.

  1. There is a manifest demand for soft actuators,[3,4] so, during last years, different 36 approaches have been made to get polymers which preserve an elastic nature in the 37 temporary shape.[5–7]. Discuss each cited reference individually.

Following recommendation of reviewer 2, each reference were briefly explained in the introduction section

  1. There is some error in the manuscript "Error! Reference source not found..'

It seems there was an error with final docx file uploaded to the platform to review. This problem has been fixed. We apologize for the inconvenience.

  1. Revise the conclusion section.

Conclusion section was rewritten following reviewer 1 and reviewer 2 recommendations.

  1. Compare the findings of current research work with past research

Shape memory elastomers using the ionic temperature as switching transition are rubbery in all the temperature range and they show a quite different properties when they are compared with other shape memory polymers where switching transition is based on Tg and/or Tm. For that reason, a quantitative comparison between SM elastomers, SM thermoplastics or SM thermoset has not sense in this work, independently of the applied stimulus. The main idea of incorporating conductive fillers were i) improve the shape memory properties of these elastomers (compared to the unfilled compounds) and ii) demonstrate that electric stimulus can be incorporated in SM elastomers, as a proof of concept for further studies

Round 2

Reviewer 1 Report

Authors answered correctly but need evidence to support your statement. Insert references to support your claim and described it inside the manuscript below statement.

“Q: In table 1 author presented CB 5-30 phr but why did authors only prepare 5-15 in the cases of CNT.

Ans: Processability of rubber compounds with MWCNT contents higher than 15 phr is an issue due to MWCNT aggregation and the high viscosity of the rubber matrix. On the other hand, the electrical conductivity of rubber compounds with 15 phr of MWCNT where similar to those obtained with higher amounts of CB”.

In keywords semicolon (;) need between two keywords instead of simple comma. Please follow journal guidelines.

References style are not the same as per journal guidelines. Polymer journal need journal abbreviations.

Why do authors put citations after a full stop. Please follow the journal guidelines.

Ref [8] has problem, so much space inside the text. Make it correct.

The First character of each word in the Title of the manuscript need to be Capital as per the journal guidelines. Check author guidelines. Check also recently published paper in the polymers journal and correct it accordingly.

One more suggestion: when authors revise any text inside the manuscript, the changes part should be highlighted or make a different colour.  

Here reviewer providing you, polymers journal references style, please check it carefully. Don’t follow other's software output styles.

References

References must be numbered in order of appearance in the text (including citations in tables and legends) and listed individually at the end of the manuscript. We recommend preparing the references with a bibliography software package, such as EndNote, ReferenceManager or Zotero to avoid typing mistakes and duplicated references. Include the digital object identifier (DOI) for all references where available.

Citations and references in the Supplementary Materials are permitted provided that they also appear in the reference list here.

In the text, reference numbers should be placed in square brackets [ ] and placed before the punctuation; for example [1], [1–3] or [1,3]. For embedded citations in the text with pagination, use both parentheses and brackets to indicate the reference number and page numbers; for example [5] (p. 10), or [6] (pp. 101–105).

  1. Author 1, A.B.; Author 2, C.D. Title of the article. Abbreviated Journal Name Year, Volume, page range.
  2. Author 1, A.; Author 2, B. Title of the chapter. In Book Title, 2nd ed.; Editor 1, A., Editor 2, B., Eds.; Publisher: Publisher Location, Country, 2007; Volume 3, pp. 154–196.
  3. Author 1, A.; Author 2, B. Book Title, 3rd ed.; Publisher: Publisher Location, Country, 2008; pp. 154–196.
  4. Author 1, A.B.; Author 2, C. Title of Unpublished Work. Abbreviated Journal Name year, phrase indicating stage of publication (submitted; accepted; in press).
  5. Author 1, A.B. (University, City, State, Country); Author 2, C. (Institute, City, State, Country). Personal communication, 2012.
  6. Author 1, A.B.; Author 2, C.D.; Author 3, E.F. Title of Presentation. In Proceedings of the Name of the Conference, Location of Conference, Country, Date of Conference (Day Month Year).
  7. Author 1, A.B. Title of Thesis. Level of Thesis, Degree-Granting University, Location of University, Date of Completion.
  8. Title of Site. Available online: URL (accessed on Day Month Year).

Author Response

Reviewer 1:

Authors answered correctly but need evidence to support your statement. Insert references to support your claim and described it inside the manuscript below statement.

“Q: In table 1 author presented CB 5-30 phr but why did authors only prepare 5-15 in the cases of CNT.

Ans: Processability of rubber compounds with MWCNT contents higher than 15 phr is an issue due to MWCNT aggregation and the high viscosity of the rubber matrix. On the other hand, the electrical conductivity of rubber compounds with 15 phr of MWCNT where similar to those obtained with higher amounts of CB”.

Following reviewer 1 recommendation, a paragraph was added in the text, both in introduction and materials sections, to clarify this fact.

In keywords semicolon (;) need between two keywords instead of simple comma. Please follow journal guidelines. Format was corrected.

References style are not the same as per journal guidelines. Polymer journal need journal abbreviations. References style was corrected with journal abbreviations.

Why do authors put citations after a full stop. Please follow the journal guidelines. Format was corrected following polymer journal references style.

Ref [8] has problem, so much space inside the text. Make it correct. Format was corrected.

The First character of each word in the Title of the manuscript need to be Capital as per the journal guidelines. Check author guidelines. Check also recently published paper in the polymers journal and correct it accordingly. Format was corrected.

One more suggestion: when authors revise any text inside the manuscript, the changes part should be highlighted or make a different colour. As stated by MDPI editor, all revisions were made using the “Track Changes” function for MS Word, such that changes can be easily viewed by the editors and reviewers.

Here reviewer providing you, polymers journal references style, please check it carefully. Don’t follow other's software output styles. Format was corrected following polymer journal references style.

Reviewer 2 Report

Now, the manuscript is improved and can be accepted for publication.

Author Response

Reviewer 2:

Now, the manuscript is improved and can be accepted for publication.

We thank reviewer 1 and 2 for all their recommendations.
